# Where the wild bees are: Birds improve indicators of bee richness

Josée S. Rousseau[1]*, Alison Johnston[1,2], Amanda D. Rodewald[1,3]

**1** Cornell Lab of Ornithology, Ithaca, New York, United States of America, **2** Centre for Research into Ecological and Environmental Modelling, School of Mathematics and Statistics, University of St. Andrews, St Andrews, United Kingdom, **3** Cornell Atkinson Center for Sustainability and Department of Natural Resources and the Environment, Cornell University, Ithaca, New York, United States of America

* jrousseau@pointblue.org

## Abstract

Widespread declines in wild bee populations necessitate urgent action, but insufficient data exist to guide conservation efforts. Addressing this data deficit, we investigated the relative performance of environmental and/or taxon-based indicators to predict wild bee richness in the eastern and central U.S. Our methodology leveraged publicly available data on bees (SCAN and GBIF data repositories), birds (eBird participatory science project) and land cover data (USDA Cropland Data Layer). We used a Bayesian variable selection algorithm to select variables that best predicted species richness of bees using two datasets: a semi-structured dataset covering a wide geographical and temporal range and a structured dataset covering a focused extent with a standardized protocol. We demonstrate that birds add value to land cover data as indicators of wild bee species richness across broad geographies, particularly when using semi-structured data. These improvements likely stem from the demonstrated sensitivity of birds to conditions thought to impact bees but that are missed by remotely sensed environmental data. Importantly, this enables estimation of bee richness in places that don't have direct observations of bees. In the case of wild bees specifically, we suggest that bird and land cover data, when combined, serve as useful indicators to guide monitoring and conservation priorities until the quality and quantity of bee data improve.

## Introduction

There is urgent need to protect declining populations of wild bees and avoid loss of the ecosystem services they provide [1–5]. Unfortunately, conservation action continues to be stymied by the paucity of rigorous information on bee populations and communities [6,7]. Despite several initiatives to fill data gaps (e.g., [8,9]) and increased numbers of observations submitted to participatory science projects like iNaturalist, data on wild bees are likely to remain deficient in the near term. As a result, many locations still lack sufficient bee data to assess their conservation value. When facing such information needs, a common approach has been to develop indicators that can be used to understand populations or communities [10–12], evaluate environmental conditions [13–16], and/or inform management [17,18]. Although the idea of indicators may seem relatively straightforward from a conceptual

**Data availability statement:** Publicly available bee datasets were analyzed in this study,

available from the following sources: 1) Chesshire, P. R., Fischer, E. E., Dowdy, N. J., Griswold, T. L., Hughes, A. C., Orr, M. C., et al. (2023). Completeness analysis for over 3000 United States bee species identifies persistent data gap. Ecography n/a, e06584. doi: 10.1111/ecog.06584, 2) GBIF.org (2022): GBIF Occurrence Download available at: https://www.gbif.org/, and 3) SCAN (2022): Available at: http://scan-bugs.org/portal/index.php. The codes used in this analysis are available at: https://zenodo.org/records/14705885. Publicly available bird datasets were also analyzed in this study, available from the eBird Basic Dataset: eBird Basic Dataset. Version: EBD_relApr-2022. 2022.

**Funding:** The research included in this paper was made possible through funding by the Walmart Foundation to JR, AR, and AJ. The findings, conclusions and recommendations presented in it are those of the authors alone, and do not necessarily reflect the opinions of the Walmart Foundation. The funders had no role in study design, data collection and analysis, decision to publish, or preparation of the manuscript.

**Competing interests:** The authors have declared that no competing interests exist.

perspective, their development is often complicated, and few studies directly compare different methodologies used to create them. Consequently, we poorly understand the effectiveness of different approaches to developing indicators and, specifically, the extent to which they are more reliably built with environmental or taxon-based variables [19,20].

Environmental indicators or surrogate measures of biodiversity can include metrics describing ecosystems or landscapes, such as vegetation indices (e.g., Normalized Difference Vegetation Index - NDVI), structural complexity, land use and land cover, or topography [15,21,22]. The underlying rationale for using environmental surrogates is that aspects such as land cover classes can reflect habitat or landscape conditions that affect species. One clear advantage of environmental surrogates is the ease with which one can access a variety of remotely sensed data representing broad spatial extents and different time periods (e.g., including infrared; [23,24]). However, the resolution and detail of remotely-sensed data are often coarse and insufficient to describe ecological attributes required by any given species. In particular, satellite imagery is unlikely to capture microhabitat features, species interactions (e.g., the presence of competitors or predators), or land management practices [24,25]. These limitations might be resolved, in part, by using data on other species that may capture multiple dimensions of habitat as well as species interactions better than environmental surrogates ([26,27], but see [20]).

Taxon-based indicators use data on a single species, an assemblage of species, or an ecological community as proxies to represent other species or indirectly describe aspects of the environment that are difficult to measure directly [18,28]. Though single species have been successfully used as proxies [29–32], indicators based on species assemblages are generally recommended and considered to perform better [33–36], especially when species collectively represent a range of life histories, habitats, and sensitivities to habitat modifications and disturbances [12,27,37]. These assemblages of individual species can represent multiple taxa, as with Management Indicator Species used by U.S. Forest Service ([38]; e.g., [39]). In contrast, indicators built from community metrics, like species richness, have had limited success at characterizing the richness of biodiversity of other taxa, and this in multiple types of land covers [40–45]. Previous research demonstrates the usefulness of indicators based on a wide range of taxa, including mammals [11,46,47], butterflies [12,48], fish [49], and birds [11,50,51]. Such taxon-based indicators are typically most effective when based upon species that are relatively common, easily detected, and cost-effective to sample [37,52]. Among animal taxa, invertebrates have been used as proxies for environmental health, especially as related to pollution [53] while birds are typically used to assess biodiversity and habitat conditions for suites of species [54–56]. Advantages of using birds is that they are common, easy to survey, many are strongly associated with habitat and landscape attributes, and they are affected by processes operating across multiple scales [37,57–59]. Moreover, the proliferation of participatory science projects like eBird, have made birds unrivaled in terms of data availability over time and space and at low cost [60–63].

Surprisingly, most indicators are constructed using *either* environmental or taxon-based data. Few examples exist for indicators that combine environmental and species indicators, despite the potential to leverage the advantages of each. Ferris and Humphrey [64] alluded to using indicator species in combination with habitat structures as 'potential indicators of biodiversity', however, to our knowledge, González et al. [17] and Fleishman et al. [27] were first to document that a combination of environmental variables and indicator species best explained restoration success and variation in species richness, respectively.

Here we investigate which combination of environmental and species data best predicts species richness of wild bees in the eastern and central U.S.. Concern about wild bees continues to rise as populations decline, species are extirpated, and key habitat resources are lost, yet data

deficiencies still limit our ability to detect and respond to changes. The convergence of urgency to act and limited data upon which to base actions makes bees a group for which indicators are likely to be valuable. Previous research used expert-identified and remotely-sensed land cover classes to indicate wild bee abundance across the U.S. [65,66]. However, many habitat resources used by bees, such flowering plants or ground characteristics [67,68], are not easily detected by satellites [25]. Likewise, a variety of stressors, including pesticides [69–71] and climate change [71,72], may not be evident from remotely-sensed data. Because many bird species are sensitive to habitat elements at multiple spatial scales (e.g., microhabitat, stand, landscape, and region; [59,73–75]) and land management practices [76,77], we hypothesized that combining bird and land cover data would best predict the resources, habitats, and landscapes that are associated with diverse bee communities. Here, we compared the performance of indicators of bee richness that were constructed from data on birds, land cover types, or a combination of both. Our intention was to develop a tool to guide monitoring, land management, and conservation efforts for bees across large spatial scales where sufficient bee data are not currently available.

## Methods

### Bee, bird, and land cover data

We used publicly available and field-based bee and bird data collected in the eastern and central regions of the US (Fig 1) to predict species richness of wild bees (thereafter bee richness). We compared the relative performance of models including predictors that were based on land cover types alone, bird data alone, or a combination of the two, using bee data from both structured and semi-structured datasets.

The structured bee dataset consists of data collected using a rigorous protocol and contains information about the bees and associated survey effort [78]. It is represented by the U.S. Geological Survey data [79], which contains protocol and effort information and can be standardized as the number of bee species per trap in each survey. We selected records of wild bees (excluding honeybees *Apis mellifera*) associated with the Bee Inventory and Monitoring Laboratory protocol, which recommends and typically used nets and 3.25 and 12 oz pan traps at each site [8]. We used surveys where at least 90% of the specimens were identified, and excluded records with missing species identification or with geographic uncertainty exceeding 3 km. We further restricted the temporal range to five years (2011–2015) and the geographical extent to a few states in eastern U.S (Fig 1). This produced a dataset with 48,654 bee records, representing 345 species and 1,583 surveys distributed across 390 3x3 km grid cells. We computed the average number of species per trap per survey for each grid cell, as our standardized measure of bee richness.

The semi-structured dataset was sourced from Chesshire et al. [80], and supplemented with 2021 records from Global Biodiversity Information Facility (GBIF; [81]) and Symbiota Collections of Arthropods Network [82]. Records were collected from 2007 to 2021 in the central and eastern U.S. using a wide range of survey methods and efforts. The 2021 supplemental data were subject to the same checks, filters, and species name validations as described in Chesshire et al. [80]. We also removed records that were duplicated, lacked species identification, location, or date, or for which uncertainty about geographic location exceeded 3km. This gave us a dataset of 476,584 bee records, representing 792 species across 26,673 3x3 km grid cells. For each grid cell, we calculated the number of species per survey, where a survey was defined by a unique combination of latitude, longitude, and date. Surveys with only one bee, as was the case for most iNaturalist submissions, were excluded as were grid cells with only one survey and fewer than 30 total bee records [83–85]. For each grid cell, we calculated the mean number of species per survey as a standardized metric of bee richness.

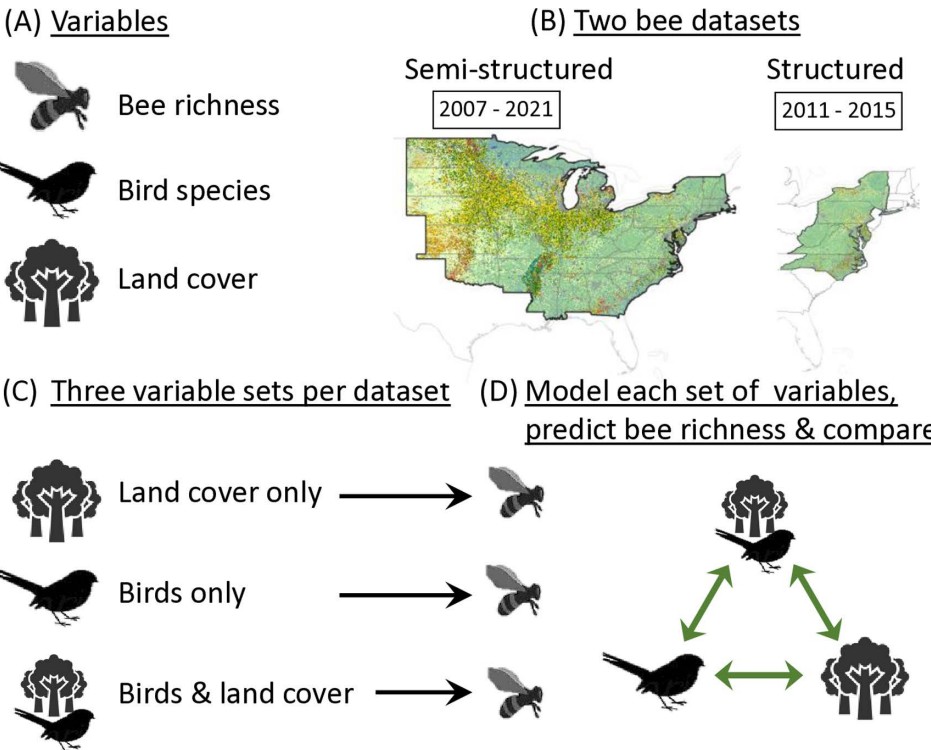

**Fig 1. Schema of our methodology.** (a) We considered bird species and land cover types, to predict bee richness. (b) We used field data from two bee datasets - a semi-structured and structured datasets and summarized them over 3x3 km grid cells, across eastern and central U.S. (c) We compared three sets of variables - land cover types only, birds only, and birds and land cover types. (d) For each set of variables, we created 100 sub-models using a Bayesian variable selection process. We considered model fit and validation to identify the set of variables that best predicted bee richness. We statistically compared model fit, for each dataset.

Bird data were extracted from the eBird Basic Dataset (EBD; [86]), which consists of bird species checklists submitted by volunteers and subsequently reviewed by experts [87,88]. Bird records were used from grid cells that also contained bee data. We selected checklists collected during the bird breeding season (May 15 to August 15) using stationary or traveling protocols lasting five to 300 minutes. We excluded surveys from observers with limited experience (i.e., fewer than three submitted checklists in our dataset) or that failed to record all species detected along with numbers of individuals for each species. Bird abundance data were standardized, within each dataset, accounting for checklist variation in survey effort, time of day, and protocol (S1). We fit a Generalized Additive Model (GAM) for each species separately using the package mgcv (version 1.9–1; [89]). The response was the species count per checklist and the predictors were: survey duration, distance traveled by the observer, time the observation started, and protocol. For each checklist and dataset, we calculated species-specific residuals on the log scale from this average relationship. This allowed us to characterize whether checklists recorded high or low species counts, accounting for the checklist effort. For each species and dataset, we then averaged across all checklists within a grid cell, to calculate a mean residual per grid cell, where a positive residual indicates grid cells where a species was more abundant than expected.

To avoid constructing an indicator based on rare species, we established prevalence thresholds to ensure that models included only those bird species that were detected in at least 20% of the grid cells, per dataset [90] and had a breeding distribution [91] covering at least 40% of

each study area. The reason for this is that rare species are (by definition) rarely observed and are therefore more likely to cause overfitting rather than true associations in our models. One exception to these thresholds were grassland obligate species, for which several were included despite being slightly below the 20% prevalence, because we were especially interested in agricultural landscapes and due to possible correlation with bee habitats. We excluded all records from bird species that are typically detected as flyovers during the breeding season (e.g., many raptors and some aerial feeders, S2) because they could not be linked to local habitat conditions. A total of 79 bird species were considered in our models to be associated with the semi-structured bee dataset and 72 to be associated with the structured dataset (S3 Table).

Land cover data were sourced from Cropland Data Layer (CDL), a geo-referenced 30-meters resolution raster, originally obtained from satellite, and categorized into crop-specific land cover types [92]. We aggregated the CDL from 120 to 45 categories relevant to bee ecology (following [66]; S4 Table) and calculated the percentage of each land cover type per 3x3 km grid cell. In our models, we considered only the most common land cover predictors that had a prevalence of at least 20% within each study area (they were present in at least 20% of our grid cells; S4 Table). We used data from the 2021 CDL in association with the semi-structured dataset analysis and from 2013 for the structured one.

## Modeling

We modeled bee richness within each 3x3 km grid cell, separately for each dataset. Modeling with the semi-structured dataset incorporated 79 bird species and 21 land cover variables from 2,585 grid cells that contained data on bees, birds, and land cover types and met our threshold for analysis. The structured dataset included 72 bird species and 20 land cover variables across 194 grid cells.

After scaling each variable, we used a Bayesian variable selection process from the R package leaps (version 3.2; [93]) to select the best predictors of bee richness using three different sets of candidate predictor variables: (a) land cover types only, (b) birds only, or (c) a combination of land cover types and birds (Fig 1). For each set of predictors we created 100 sub-models, each limited to 10 predictors because model fit did not meaningfully improve when >10 were included. We calculated the predicted bee richness within each grid cell by averaging the bee richness predictions from these 100 sub-model predictions. This created three modeled predictions of bee richness, each created with a different set of candidate predictor variables. Sub-models with a large number of potential predictors took a prohibitively long time to run. Therefore, we used a variable subset selection process to empirically remove predictors that were unlikely to be useful in predicting bee richness (S5). We conducted a sensitivity analysis to ensure that this variable selection process would not impact the results (S5). We also assessed the presence of multicollinearity among predictors in each sub-model using Variance Inflation Factors (VIF). No sub-models had predictors with a VIF of 5 or higher, suggesting minimal multicollinearity [94].

We assessed the accuracy of our three models separately for each dataset using a five-fold cross-validation process. We re-ran the modeling procedure with each of five subsets of 80% of the data, each time creating 100 sub-models and averaging over sub-model predictions to produce modeled predictions for each grid cell within the 20% validation data. This ensured that we were assessing predictions using independent data to prevent positive conclusions being driven by overfitting to the modeled data. We repeated this process five times to create predictions for every grid cell in the original dataset. We repeated the whole procedure for models constructed from each of the three sets of predictor variables: (a) land cover types only, (b) birds only, or (c) a combination of land cover types and birds.

We compared observed to predicted values of bee richness within each grid cell (S6 Fig) using a correlation coefficient. We statistically compared the correlation coefficients between observed and predicted bee richness, for models constructed from each of the three sets of predictor variables – land cover types only, birds only, birds and land cover types. We used the correlation coefficient tests proposed by Hittner et al. [95] and available through the R package cocor (version 1.1–4; [96]), to determine which set of variables best predicted observed bee richness, within each dataset.

The best set of variables was used to predict bee richness across the study area associated with each dataset. In order to do this, we needed estimates of relative bird abundance in all grid cells, not only those with eBird checklists, therefore we used the estimated relative abundance per species per grid cell from eBird Status Data Products, which are standardized and calculated using ensemble models [97,98]. The bee richness point estimates and associated map represent the mean prediction from the 100 sub-models, at each location. For the bee richness uncertainty from the model selection process, we calculated a 90% confidence interval of the bee richness predictions from the 100 sub-models, at each location. Since the bee richness values are relative, the confidence interval range was normalized by the range of point estimates across the entire study extent. This scaled uncertainty reflects the percentage of the variation across models at a given location, relative to the full range of variation across point estimates within all locations.

## Results

The combination of bird and land cover data yielded the most accurate predictions of bee richness using either the structured data (model fit $R^2$ = 0.28, validation $R^2$ = 0.21, n = 194) and the semi structured data (Table 1; semi-structured data model fit $R^2$ = 0.14, validation $R^2$ = 0.14, n = 2585 grid cells). Plots of observed and predicted values for these analyses are available in the supplementary material (S6 Fig).

The inclusion of both land cover types and bird species significantly improved correlation coefficients by >15% in the structured dataset and 3% in the semi-structured dataset compared to using either land cover types or birds alone. These improvements were statistically significant for the semi-structured dataset, with the model with both land cover types and birds being better than land cover types only (p < 0.001) and birds only (p < 0.001). For the structured dataset, the model with both land cover types and birds was significantly better than the model with land cover types only (p = 0.007), but did not show a significant improvement over the model with birds only (p = 0.35). Model fit for birds and land cover types was twice as good and significantly improved (p = 0.01) using the structured dataset compared with semi-structured dataset (Table 1).

Table 1. **Comparison of model fit and associated five-fold validation coefficient of determination among the three sets of variables: land cover types only, birds only, and birds & land cover types. The model fits are compared among two datasets: data covering the midwest and eastern USA and subset of high quality data covering eastern USA.**

| Region & dataset | Model fits | Land cover types only | Birds only | Birds & land cover types |
|---|---|---|---|---|
| Semi-structured dataset | Model fit - R2 | 0.11 | 0.11 | 0.14 |
| Semi-structured dataset | 5-fold validation - R2 | 0.11 | 0.10 | 0.14 |
| Structured dataset | Model fit - R2 | 0.12 | 0.24 | 0.28 |
| Structured dataset | 5-fold validation - R2 | -0.005 | 0.14 | 0.21 |

**Table 2. List of predictors selected at least once in each of the 100 models per variables set - land cover types only, birds only, and birds and land cover types - and their associated mean estimate, standard deviation in the estimates, and number of models in which they were selected. Predictor variables are sorted according to frequency of inclusion in "birds & land cover types" models. Scientific names of bird species are included in S3 Table.**

| Predictor variables | Mean of estimates, SD (# of models) | | |
|---|---|---|---|
| | Land cover types only | Birds only | Birds & land cover types |
| Intercept | 4.76, 0.00 (100) | 4.76, 0.00 (100) | 4.76, 0.00 (100) |
| Double crop | 0.51, 0.03 (100) | . | 0.37, 0.01 (100) |
| Urban low-density | -0.79, 0.15 (100) | . | -0.55, 0.05 (100) |
| Barren | 0.48, 0.06 (98) | . | 0.60, 0.02 (100) |
| Deciduous forest | 0.47, 0.09 (63) | . | 0.60, 0.05 (100) |
| Carolina Wren | . | 0.85, 0.08 (100) | 0.64, 0.09 (100) |
| Idle cropland | 0.35, 0.04 (81) | . | 0.31, 0.04 (72) |
| Common Yellowthroat | . | -0.37, 0.04 (68) | -0.36, 0.06 (67) |
| Black-capped Chickadee | . | -0.37, 0.05 (49) | -0.40, 0.04 (60) |
| Blue Jay | . | -0.46, 0.05 (95) | -0.36, 0.05 (51) |
| Gray Catbird | . | 0.42, 0.05 (90) | 0.32, 0.04 (45) |
| Ruby-throated Hummingbird | . | -0.26, 0.04 (8) | -0.28, 0.02 (41) |
| Alfalfa | -0.39, 0.05 (77) | . | -0.29, 0.02 (37) |
| Open water | -0.39, 0.06 (100) | . | -0.27, 0.02 (24) |
| Scarlet Tanager | . | 0.48, 0.08 (90) | 0.29, 0.03 (23) |
| Warbling Vireo | . | -0.41, 0.04 (95) | -0.30, 0.03 (20) |
| Dickcissel | . | -0.29, 0.03 (36) | -0.27, 0.02 (18) |
| Orchard Oriole | . | 0.35, 0.04 (78) | 0.27, 0.02 (14) |
| European Starling | . | . | 0.25, 0.02 (6) |
| Grass pasture | 0.10, 0.08 (6) | . | 0.24, 0.02 (5) |
| Chipping Sparrow | . | 0.33, 0.04 (51) | 0.24, 0.01 (3) |
| American Crow | . | 0.26, 0.02 (12) | 0.24, 0.01 (3) |
| Red-bellied Woodpecker | . | -0.31, 0.06 (2) | -0.34, 0.03 (3) |
| Rose-breasted Grosbeak | . | -0.18 (1) | -0.27, 0.03 (2) |
| Corn | -0.47, 0.14 (58) | . | -0.27 (1) |
| Northern Cardinal | . | -0.61, 0.07 (100) | -0.33 (1) |
| White-breasted Nuthatch | . | -0.31, 0.03 (10) | -0.32 (1) |
| Downy Woodpecker | . | -0.31, 0.01 (8) | -0.26 (1) |
| American Redstart | . | -0.24, 0.02 (7) | -0.29 (1) |
| Yellow-throated Vireo | . | . | -0.26 (1) |
| Mixed forest | -0.39, 0.07 (87) | . | . |
| Urban high-density | -0.38, 0.11 (53) | . | . |
| Coniferous forest | -0.33, 0.09 (45) | . | . |
| Herbaceous wetland | -0.29, 0.07 (36) | . | . |
| Developed open space | 0.27, 0.03 (20) | . | . |
| Urban medium-density | -0.39, 0.14 (20) | . | . |
| Bean | -0.31, 0.12 (18) | . | . |
| Grain | -0.22, 0.04 (18) | . | . |
| Woody wetland | -0.09, 0.08 (7) | . | . |
| Shrubland | 0.09, 0.01 (5) | . | . |
| Grass | 0.01, 0.01 (4) | . | . |
| Orchard | 0.03, 0.01 (4) | . | . |
| Wood Thrush | . | 0.37, 0.07 (36) | . |

*(Continued)*

**Table 2.** (Continued)

| Predictor variables | Mean of estimates, SD (# of models) | | |
| --- | --- | --- | --- |
| | Land cover types only | Birds only | Birds & land cover types |
| Cliff Swallow | . | 0.26, 0.02 (29) | . |
| American Robin | . | -0.37, 0.10 (11) | . |
| Yellow-billed Cuckoo | . | 0.27, 0.02 (11) | . |
| Blue Grosbeak | . | 0.29, 0.05 (6) | . |
| Song Sparrow | . | -0.28, 0.02 (4) | . |
| American Goldfinch | . | -0.22 (1) | . |
| Brown Thrasher | . | 0.17 (1) | . |
| House Sparrow | . | -0.25 (1) | . |

Focusing on the models with both birds and land cover predictors, the semi-structured and structured datasets had 9 and 5 land cover variables, respectively, and 20 and 26 birds that were selected in at least one of the 100 sub-models (Table 2 and S7 Table).

Those land cover types and bird variables that were selected in all 100 sub-models typically had a large effect size, based on their mean coefficients. Five variables were selected within all 100 sub-models using the semi-structured dataset — deciduous forest, barren land, double crop, low-density urban, and Carolina Wren — and two with the structured dataset, grain and Gray Catbird (Table 2 and S7 Table). With the exception of low-density urban landscapes, the most selected variables were positively correlated with bee richness (Table 2; S7 Table).

Our indicator predicted that bee richness in the eastern and central U.S. was generally higher on the East Coast along the Appalachian Mountains and lower in the Midwest, particularly around Iowa (Fig 2A). Uncertainty in predicted bee richness was lowest around North and South Dakota, Illinois, and along the Atlantic coast, while it was highest near West Virginia, eastern Kentucky, and southern Missouri (Fig 2B).

## Discussion

Tools for guiding the conservation of data-deficient taxa often include environmental or taxon-based indicators. Though only one type of variable typically is used to create an indicator (but see [17,27]), our results indicate that combining data from both the environment and other taxa may significantly improve the prediction accuracy and, thus, may better inform conservation actions. Unlike previous work that relied upon land cover data to predict bee abundance [65,66], we found that bird data added value over land cover types alone and improved our ability to predict species richness of wild bees. The usefulness of birds is not surprising, given that they are known to be effective indicators for ecological conditions [14,99,100] and other taxa, such as rodents and butterflies [11,12,27,101,102].

Several factors may explain why the combination of birds and land cover variables predicted bee richness better than using either land cover types or birds alone. First, bird and land cover variables likely offer complementary insights into habitat quality. A broad category of land cover, such as 'deciduous forest', usually includes a wide range of floristic composition, habitat structure, patch configuration, age, and land management practices [103–106]. For example, numbers of flowering plants that attract bees are often greater in early-successional than mature forests [103]. Likewise, forests in which understories were replaced by manicured lawn (e.g., wooded parks) provide less floral resources to bees [107]. In such cases, the presence or abundance of particular bird species (e.g., open woodland species like Chipping

(A)

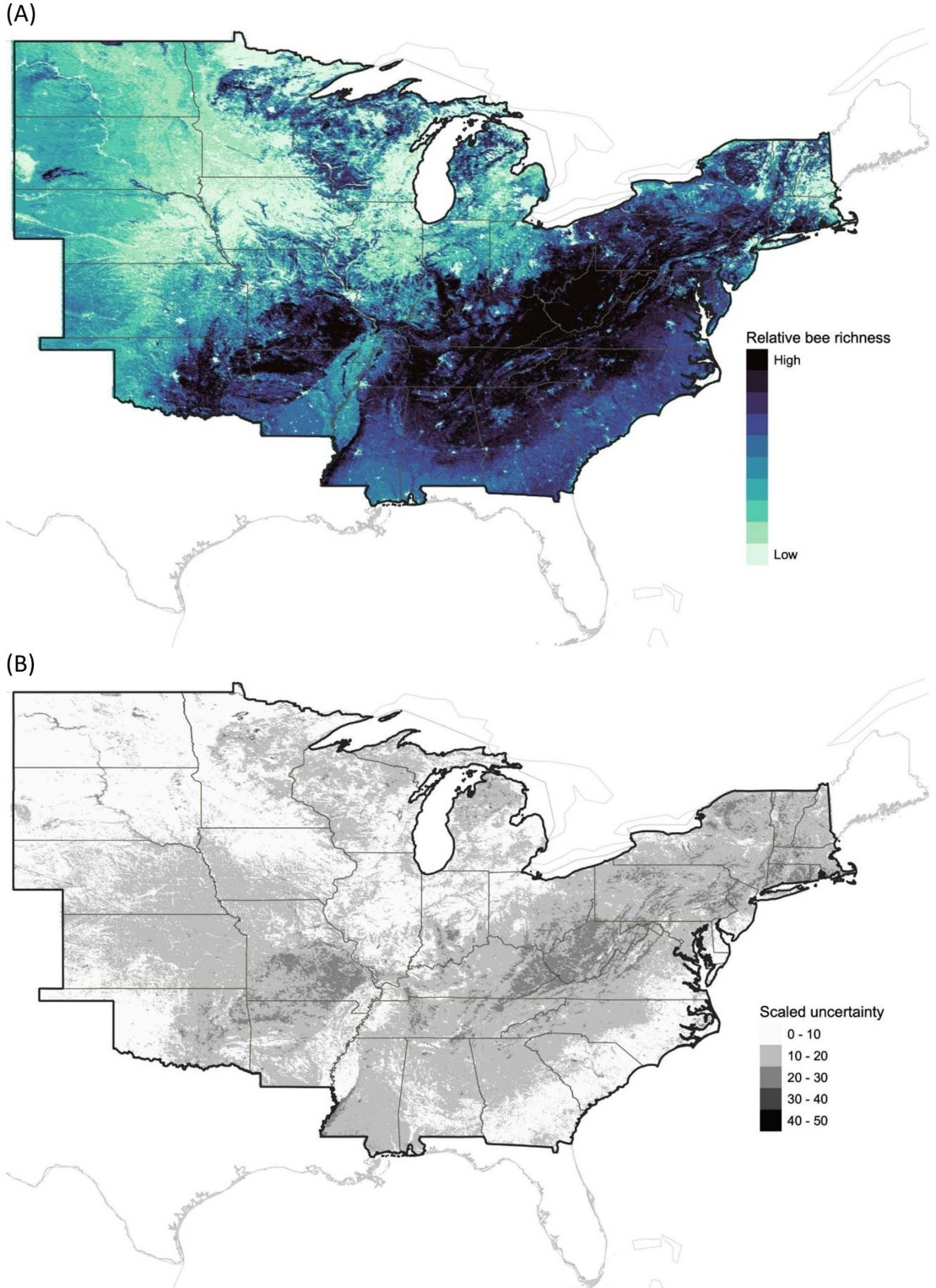

(B)

**Fig 2. Predicted bee richness using birds and land cover variables.** (a) The dark blue represents locations where a relatively higher bee richness is expected, compared to other locations within our study extent. (b) The scaled uncertainty associated with bee richness for each grid cell, where dark color represents higher uncertainty. These high uncertainty locations had a high percentage of variation across all models relative to the full range of variation across point estimates within all locations.

Sparrow (*Spizella passerina*), forest-understory species like American Redstart (*Setophaga ruticilla*) or Wood Thrush (*Hylocichla mustelina*)) can provide additional information to better identify habitats favored by bees. Indeed, the presence of fruit-eating bird species, such as Gray Catbird (*Dumetella carolinensis*), as predictors highlight the importance of forests containing flowering shrubs and trees for bee communities [108,109]. Second, individual land cover variables alone may not adequately capture the influence of habitat juxtaposition or the co-occurrence of different habitats in close proximity. For example, while 'double crops'—the practice of growing a winter and summer crop on the same land—may provide sufficient flower resources, bees may also need easy access to nearby, less disturbed habitats for nesting. Such habitats might include 'idle cropland', 'grass pastures', or 'deciduous forests', which are often better indicated by bird species that nest in trees but forage in open areas. Third, birds and bees select their habitat based on resources available across multiple scales [110–115]. As such, birds are likely to incorporate multi-scale information relevant to bee populations that would not be available through land cover types alone. For instance, the presence of species like Orchard Oriole (*Icterus spurius*) can signal the availability of open woodlands, orchards, woody hedgerows, and flowering plants used for nesting and foraging resources by orioles and wild bees alike. Fourth, the combined bird and land cover variables selected as predictors represent a wide range of bee habitats, which suggest that a higher bee richness may be associated with heterogeneous landscapes [116–118] that include multiple types of nesting substrates to accommodate ground and cavity nesting bee species and a diversity of flower resources, from crops, shrubs, and trees.

The geographic pattern of wild bee richness in the eastern half of the U.S., as predicted by our indicators (Fig 2A), is consistent with previous reports that wild bee richness and/or abundance is highest in landscapes characterized by a mosaic of deciduous forest and low-intensity agriculture and lowest in areas dominated by intensive agriculture, such as the Midwest (Fig 2; [66,69,80]). Consistent with the results from Koh et al., [66], we found high levels of uncertainty in our predictions in certain regions, particularly in areas like the Appalachian and Ozark Mountains (Fig 2B). Given that many bee species require forest habitats during particular life stages [104,119–121], we were not surprised to find large positive effects of deciduous forests and Carolina Wrens, a bird species associated with gaps in deciduous forest [122]. Also unsurprising were the negative associations we detected between bee richness and intensive agricultural crops, such as corn and alfalfa. Corn monocultures are known to have low bee richness [123], in part because of intensive management practices like tilling and pesticide applications, whereas alfalfa is used by relatively few bee genera [124]. Importantly, we recognize that species richness does not necessarily indicate conservation value. High species richness could result from communities comprised mainly of generalists and common species, whereas areas of low richness may be home to specialized and rare species might warrant more conservation attention [6,7,125,126]. For these reasons, establishing conservation priorities is best done in consultation with experts or, ideally, after ground-truthing with field surveys.

Using one taxon as an indicator for another requires consideration of ecological context, such as threats affecting both groups, species interactions, and the spatial and temporal scales at which they utilize their habitat. In our case, the breeding territory size and season of most birds align well with the timing and habitat size requirements of many bees. That said, we recognize that bees may require unique resources. For instance, ground-nesting bees may exhibit preferences for specific below-ground resources [68] that may not be well indicated by birds. Additionally, bees are likely influenced by micro-habitats at a finer scale than birds, such as the availability of small bare ground patches. Lastly, the breeding season of birds may include different density-dependent processes compared to bees, where a higher abundance of birds

is not always correlated with higher habitat quality [54]. While the association of certain bird species with bee richness may be intuitive, including all bird species *a priori* in our analysis provided insights on novel relationships between these birds, bee richness, and the habitat they occupy.

Our findings also provide insight into the influence of structured versus semi-structured data on results. The improved predictions we generated using the structured dataset are likely due to differences in data quality and scale compared to the semi-structured dataset. The structured dataset included protocol and effort information, enabling us to generate more precise bee richness estimates across space [127,128]. Additionally, the use of a limited number of years in the structured dataset minimizes variation in bee and bird species detection due to temporal changes in climate or land cover types. The comparatively narrow geographic scope of the structured dataset likely resulted in more consistent species-habitat associations across the study area and, consequently, improved model fits [129,130]. Focusing on a smaller geographical area also increased the likelihood of more bird species having their breeding distribution covering larger portions of the study area. This may be a reason the birds-only model performed relatively better using the structured than semi-structured dataset. Lastly, the semi-structured dataset may have produced a lower model fit due to its larger sample size and the broader region it covered, where many areas lacked bee data, weakening the overall results.

## Conclusions

Recent drastic declines in insect biodiversity [131–133], underscore a need to use all available information to conserve these data-deficient taxa. Despite increases in data availability from sources like satellites or participatory science projects, few have investigated the extent to which integrating data sources may improve the usefulness of indicators of taxonomic groups with limited data. We demonstrated that by combining multiple tools, we can achieve better estimates of bee richness, which are a data-deficient taxa, but also provide vital ecosystem services. While we recognize that any indicator – no matter how computationally rigorous it may be – will fall short of the value of actual field data on bees, we suggest that indicators can be useful for identifying geographic areas that may be important for bees, informing conservation efforts, and supporting monitoring programs [134].

## Supporting information

**S1 File. Standardizing bird abundance using residuals.**
(PDF)

**S2 File. List of bird species excluded from analysis.**
(PDF)

**S3 Table. Bird species used in semi-structured and structured dataset analysis.** List of bird species and their scientific names, considered in the analysis. Only bird species with a prevalence (Prev) of 20% or greater and a breeding distribution covering at least 40% of the respective study region were included: the semi-structured dataset encompassed the eastern half of the U.S., while the structured dataset focused on several eastern states. Bird species selected in at least one of the 100 models for both dataset analyses are listed.
(PDF)

**S4 Table. Land cover types used in semi-structured and structured dataset analysis.** List of land cover types considered in the analysis. Only land cover types covering 20% or more of their respective study region were included (Prev): the semi-structured dataset encompassed

the eastern half of the U.S., while the structured dataset focused on several eastern states. Land cover types selected in at least one of the 100 models for both dataset analyses are listed.
(PDF)

**S5 File. Two-step process to select relevant predictors of bee richness.** Description of the process used. Includes comparison of the number of predictors used, resulting model fit and validation r-squared, predictors selected, and associated estimates per methodology tested.
(PDF)

**S6 Fig. Plots of the observed and predicted bee richness for the semi-structured and structured analysis using land cover types and bird data model.** Includes two plots: 1) Correlation between observed and predicted bee richness using the semi-structured dataset for 2,501 locations across the eastern half of the U.S., 2007–2021. The yellow contour lines represent the density of points and the red line the linear correlation, 2) Correlation between observed and predicted bee richness using the structured dataset for 194 locations across the eastern U.S., 2011–2015. The blue contour lines represent the density of points and the red line the linear correlation.
(PDF)

**S7 Table. Structured dataset results.** List of predictors selected at least once in each of the 100 models per variables set - land cover types only, birds only, and birds and land cover types - and their associated mean estimate, standard deviation in the estimates, and number of models in which they were selected. Predictor variables are sorted according to frequency of inclusion in "birds & land cover" models.
(PDF)

## Acknowledgments

We thank P. Chesshire for sharing information about the quality control and filters that were used when supplementing her compiled dataset and to the following data owners, which provided at least 5% of the records used in the semi-structured dataset: American Museum of Natural History, iNaturalist, University of Kansas Biodiversity Institute, U.S. Department of Agriculture, and U.S. Geological Survey. Thank you to S. Droege for sharing the U.S. Geological Survey data and providing guidance on extracting records associated with the standardized protocol. We thank the following bee taxonomists who identified more than 5% of the bee records in the compiled dataset: John S. Ascher, Sam Droege, Terry L. Griswold, Wallace E. LaBerge, and D.W. Ribble. We appreciate the generous feedback from B. Danforth, the guidance provided by Matt Strimas-Mackey on accessing and manipulating eBird data, and the valuable input from colleagues in the Center for Avian Population Studies at the Cornell Lab of Ornithology that improved our work. Finally, we are especially grateful to our partners at the Cornell Atkinson Center for Sustainability, particularly Patrick Beary and Gail Phillips, for their collaboration, support, and input throughout the project.

## Author contributions

**Conceptualization:** Josée S. Rousseau, Alison Johnston, Amanda D. Rodewald.

**Data curation:** Josée S. Rousseau.

**Formal analysis:** Josée S. Rousseau.

**Funding acquisition:** Alison Johnston, Amanda D. Rodewald.

**Methodology:** Josée S. Rousseau, Alison Johnston, Amanda D. Rodewald.

**Project administration:** Amanda D. Rodewald.

**Software:** Josée S. Rousseau.

**Supervision:** Alison Johnston, Amanda D. Rodewald.

**Visualization:** Josée S. Rousseau.

**Writing – original draft:** Josée S. Rousseau.

**Writing – review & editing:** Josée S. Rousseau, Alison Johnston, Amanda D. Rodewald.

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
