## [Decision Letter · Decision Letter 0]

3 Jan 2025

PONE-D-24-49004Where the wild bees are: birds improve indicators of bee richnessPLOS ONE

Dear Dr. Rousseau,

Thank you for submitting your manuscript to PLOS ONE. After careful consideration, we feel that it has merit but does not fully meet PLOS ONE’s publication criteria as it currently stands. Therefore, we invite you to submit a revised version of the manuscript that addresses the points raised during the review process.

Your manuscript has been revised by two reviewers that provide feedback to improve their clarity and quality. Reviewers highlight the importance and relevance of the study, but they also identify some issues which require attention. Overall, these issues should be feasible to address through careful revision.

We look forward to receiving your revised manuscript.

Kind regards,

Vicente Martínez López

Academic Editor

PLOS ONE

Journal Requirements:

2. Thank you for stating the following financial disclosure: The author(s) declare financial support was received for the research, authorship, and/or publication of this article. The research included in this paper was made possible through funding by the Walmart Foundation to JR, AR, and AJ. The findings, conclusions and recommendations presented in it are those of the authors alone, and do not necessarily reflect the opinions of the Walmart Foundation.

3. We note that Figures 1 and 2 in your submission contain [map/satellite] images which may be copyrighted. All PLOS content is published under the Creative Commons Attribution License (CC BY 4.0), which means that the manuscript, images, and Supporting Information files will be freely available online, and any third party is permitted to access, download, copy, distribute, and use these materials in any way, even commercially, with proper attribution. For these reasons, we cannot publish previously copyrighted maps or satellite images created using proprietary data, such as Google software (Google Maps, Street View, and Earth). For more information, see our copyright guidelines: http://journals.plos.org/plosone/s/licenses-and-copyright.

a. You may seek permission from the original copyright holder of Figures 1 and 2 to publish the content specifically under the CC BY 4.0 license.  

Reviewers' comments:

Reviewer's Responses to Questions

**Comments to the Author**

1. Is the manuscript technically sound, and do the data support the conclusions?

Reviewer #1: Yes

Reviewer #2: Yes

2. Has the statistical analysis been performed appropriately and rigorously? 

Reviewer #1: I Don't Know

Reviewer #2: Yes

3. Have the authors made all data underlying the findings in their manuscript fully available?

Reviewer #1: Yes

Reviewer #2: Yes

4. Is the manuscript presented in an intelligible fashion and written in standard English?

Reviewer #1: Yes

Reviewer #2: Yes

5. Review Comments to the Author

Reviewer #1: Revision: Where the wild bees are: birds improve indicators of bee richness

This manuscript investigates the performance of environmental and bird-based indicators in predicting wild bee richness in the U.S. The authors provide robust evidence that birds enhance the predictive power of land cover data as indicators of wild bee richness. The manuscript explores a novel and interesting approach (I was not familiar with) that is highly relevant to a broad audience, particularly conservationists focused on wild bees. Given that bird occurrence and diversity data are often far more extensive than comparable data for wild bees, this study addresses a significant knowledge gap.

The text is well-structured and well-written, with a particularly engaging introduction. The discussion is nuanced, effectively highlighting the strengths of the study while addressing its limitations. I support this manuscript for publication, subject to addressing the minor comments detailed below.

Abstract

• Line 23: “Widespread declines in wild bee populations necessitates”—either remove the “s” from “declines” or from “necessitates” to correct the subject-verb agreement.

Introduction

• Line 47: A closing parenthesis appears to be missing.

• Line 55: There is a double space between “them.” and “Consequently.”

• Lines 80–82: The distinction between ‘environmental health’ and ‘environmental quality’ is unclear. Are there rigorous definitions that differentiate these terms? Additionally, the statement that invertebrates are better proxies for one and birds for the other requires more explanation or supporting references.

• Lines 86–93: These lines, discussing the use of single species or assemblages, directly relate to concepts introduced in lines 74–76. Consider merging lines 74–76 with lines 86–93 for a more cohesive discussion, and then proceed with lines 76–86, which focus on specific taxa and their relevance.

• Lines 113–114: There seems to be another missing parenthesis in this section.

Methods

• Lines 149–151: I am not sure how bee richness was calculated. Specifically, how did you integrate data from net sampling when averaging the number of species per trap per survey? Further clarification on this methodology would be helpful.

• Lines 164–165: This might be a naïve observation, but in this semi-structured dataset, surveys appear to follow different protocols. Couldn’t this lead to significant variations in the number of species per survey, not due to actual differences in species richness, but as a result of methodological discrepancies? For instance, depending on the sampling method (e.g., nets, traps, or nets + traps) or the sampling duration, the number of species recorded at the same site on the same day can vary substantially; it can easily double or triple. How do you think this variability might influence the trends observed in your study? Additionally, why did you use the mean instead of the median to summarise species richness? Wouldn’t the median reduce the influence of outliers? Alternatively, why not consider the total number of species observed across all surveys within a grid cell?

• Line 170: Why did you not include pre-nuptial migration periods, such as April, in your bird-based index? April is a critical period for many early-season bee species, and excluding it might overlook their habitat requirements. Including data from pre-nuptial migration sites and their associated bird diversity could potentially improve the index's applicability to early-season bees. While, of course, re-extracting bird data is beyond the scope of this study, do you think this aspect is worth exploring in future research?

Discussion

• Line 333: It seems that a track change was not removed, specifically in the space between “bees” and (101).

Figures

• Figure 2: In the figure legend, could you clarify what “scaled uncertainty” represents? Is it expressed as a percentage or as a number of species?

• Figure 2: It might be valuable to include a map showing actual bee richness based on existing data. This would allow for a direct comparison between observed richness and the predicted values, providing additional context and insight into the model's accuracy.

Thanks for this great paper, as a bee researcher, and a bird lover, I particularly enjoyed it !

Maxence Gérard

Reviewer #2: The article presents an interesting case of searching for an improved performance of land use predictors, when combined with citizen science bird databases, to indicate species richness of wild bees. It's aim is using the former as indicators of the less know biodiversity of the latter.

In my opinion, the research has a good design and a solid methodological approach that makes its publication highly recommended.

There are only a few suggestions on how to express certain issues included in the introduction, discussion and conclusions, which I propose to the authors for their consideration. These questions relate more to the wording of these three sections and possible improvements in the formulation and interpretation of the hypotheses and results, than to methodological issues that, as I have indicated, are adequately described and justified in their choice and application.

1) The statement (lines 99 – 101) “however, to our knowledge, González et al. (17) and Fleishman et al. (27) were first to document that a combination of environmental variables and indicator species best explained restoration success and variation in species richness, respectively”, is a bit confusing. Since it is unlikely that these are the only that have adopted this approach (as one might think), it would be convenient to cite at this point some additional references published after them that can be considered background to the present research.

2) If anything, in relation to the methodology, I would appreciate an explanation in some detail of the methodological basis of the eBird product cited (reference 92, Fink et al., 2021: Lines 250-252 “we needed estimates of relative bird abundance in all grid cells, not only those with eBird checklists, therefore we used the estimated relative abundance per species per grid cell from eBird Status Data Products (92)”. Although this product is explained in the link associated to the reference, it seems appropriate that the authors (and not the reader) be the ones to extract -and include in the text- this necessary methodological note.

3) Lines 350-350: The sentence "Fourth, the combined bird and land cover variables selected as predictors represent a wide range of habitats, which suggest that a higher bee richness may be associated with heterogeneous landscapes (110–112) that include multiple types of nesting substrates to accommodate ground and cavity nesters and a diversity of flower resources, from crops, shrubs, and trees", is also a bit confusing since the argument combines nesting substrates (¿for birds?) and trophic resources (for bees? for both?). The relationship of birds with heterogeneous landscapes and structurally diverse habitats, and their consequent indicator value for habitat diversity for pollinators, should probably be differentiated. Likewise, it should be accompanied by some references that support the first relationship, since all those cited (110-112) refer exclusively to the habitat relationships of pollinators.

On the contrary, in the sentence between lines 392-394, which refers entirely to pollinators, "The structured dataset included protocol and effort information, enabling us to generate more precise bee richness estimates across space (121,122)", the second reference cited is on birds. It is not clear to me what justifies its use.

4) The article goes into little detail about the cautions to be taken when choosing birds as surrogate for invertebrates (wild pollinators in this case), especially taking into account the extensive literature on the lack of correspondence between protected areas designated with criteria based on each of these taxocenoses. For example, the following works or others with a similar approach can be cited:

- Guareschi, S., Abellán, P., Laini, A., Green, A. J., Sánchez-Zapata, J. A., Velasco, J., & Millán, A. (2015). Cross-taxon congruence in wetlands: assessing the value of waterbirds as surrogates of macroinvertebrate biodiversity in Mediterranean Ramsar sites. Ecological Indicators, 49, 204-215.

- Oberprieler, S. K., Andersen, A. N., Gillespie, G. R., & Einoder, L. D. (2019). Vertebrates are poor umbrellas for invertebrates: cross‐taxon congruence in an Australian tropical savanna. Ecosphere, 10(6), e02755.

- Rooney, R. C., & Bayley, S. E. (2012). Community congruence of plants, invertebrates and birds in natural and constructed shallow open-water wetlands: do we need to monitor multiple assemblages?. Ecological Indicators, 20, 42-50. (This last article raises as a final remark that cross-congruence between taxocenoses may be reduced in degraded ecosystems with respect to those that maintain a good level of conservation, which may also be a relevant aspect to consider here)

5) The conclusions section is this respect somewhat vague; it should specify a little more what type of conservation measures are recommended, since the selection of areas to be protected does not guarantee a great concordance between those aimed at birds and those that can protect other taxa.

In particular, studies have shown a mismatch between existing protected areas (essentially designated on the basis of plant or vertebrate taxa) and areas of modelled high diversity of wild bees (e.g. Casanelles‐Abella, J., Fontana, S., Meier, E., Moretti, M., & Fournier, B. (2023). Spatial mismatch between wild bee diversity hotspots and protected areas. Conservation Biology, 37(4), e14082.)

6) Moreover, the claim that birds can help identify areas rich in wild bee species through their indication of habitat diversity could be criticized, at least partly, on the basis of other research not reported by the authors, e.g. Geue JC, Rotter PJ, Gross C, Benkő Z, Kovács I, Fântână C, et al. (2022) Limited reciprocal surrogacy of bird and habitat diversity and inconsistencies in their representation in Romanian protected areas. PLoS ONE 17(2): e0251950. https://doi.org/10.1371/journal.pone.0251950

In addition, i found thes minor observations/corrections referring to specific lines of the manuscript:

Lines 330-333: “For example, numbers of flowering plants that attract bees are often greater in early-successional than mature forests(97). Likewise, forests in which understories were replaced by manicured lawn (e.g., wooded parks) provide less floral resources to bees_(101).

- There is a space missing between "mature forest" and "(97)"

- The space between "...floral resources to bees" and "(101)" does not need to be underlined

6. PLOS authors have the option to publish the peer review history of their article (what does this mean? ). If published, this will include your full peer review and any attached files.

**Do you want your identity to be public for this peer review?** For information about this choice, including consent withdrawal, please see our Privacy Policy .

Reviewer #1: **Yes: ** Maxence Gérard

Reviewer #2: No

---

## [Author Response · Author response to Decision Letter 0]

20 Jan 2025

Comments from the editors

• 1. Please ensure that your manuscript meets PLOS ONE's style requirements, including those for file naming. The PLOSONE style templates can be found at

o https://journals.plos.org/plosone/s/file?id=wjVg/PLOSOne_formatting_sample_main_body.pdf and

o https://journals.plos.org/plosone/s/file?id=ba62/PLOSOne_formatting_sample_title_authors_affiliations.pdf

- Done

• 2. Thank you for stating the following financial disclosure: The author(s) declare financial support was received for the research, authorship, and/or publication of this article. The research included in this paper was made possible through funding by the Walmart Foundation to JR, AR, and AJ. The findings, conclusions and recommendations presented in it are those of the authors alone, and do not necessarily reflect the opinions of the Walmart Foundation. Please state what role the funders took in the study. If the funders had no role, please state: ""The funders had no role in study design, data collection and analysis, decision to publish, or preparation of the manuscript."" If this statement is not correct you must amend it as needed. Please include this amended Role of Funder statement in your cover letter; we will change the online submission form on your behalf.

- We would like to update our financial disclosure, adding the suggested sentence. The updated financial disclosure should read: “The research included in this paper was made possible through funding by the Walmart Foundation to JR, AR, and AJ. The findings, conclusions and recommendations presented in it are those of the authors alone, and do not necessarily reflect the opinions of the Walmart Foundation. The funders had no role in study design, data collection and analysis, decision to publish, or preparation of the manuscript.”

• 3. We note that Figures 1 and 2 in your submission contain [map/satellite] images which may be copyrighted. All PLOS content is published under the Creative Commons Attribution License (CC BY 4.0), which means that the manuscript, images, and Supporting Information files will be freely available online, and any third party is permitted to access, download, copy, distribute, and use these materials in any way, even commercially, with proper attribution. For these reasons, we cannot publish previously copyrighted maps or satellite images created using proprietary data, such asGoogle software (Google Maps, Street View, and Earth). For more information, see our copyright guidelines: http://journals.plos.org/plosone/s/licenses-and-copyright.

o a. You may seek permission from the original copyright holder of Figures 1 and 2 to publish the content specifically under the CC BY 4.0 license.

We recommend that you contact the original copyright holder with the Content Permission Form (http://journals.plos.org/plosone/s/file?id=7c09/content-permission-form.pdf) and the following text: “I request permission for the open-access journal PLOS ONE to publish XXX under the Creative Commons Attribution License (CCAL) CC BY 4.0 (http://creativecommons.org/licenses/by/4.0/). Please be aware that this license allows unrestricted use and distribution, even commercially, by third parties. Please reply and provide explicit written permission to publish XXX under a CC BY license and complete the attached form.”

Please upload the completed Content Permission Form or other proof of granted permissions as an ""Other"" file with your submission. In the figure caption of the copyrighted figure, please include the following text: “Reprinted from [ref] under a CC BY license, with permission from [name of publisher], original copyright [original copyright year].”

o b. If you are unable to obtain permission from the original copyright holder to publish these figures under the CC BY 4.0license or if the copyright holder’s requirements are incompatible with the CC BY 4.0 license, please either i) remove the figure or ii) supply a replacement figure that complies with the CC BY 4.0 license. Please check copyright information on all replacement figures and update the figure caption with source information. If applicable, please specify in the figure caption text when a figure is similar but not identical to the original image and is therefore for illustrative purposes only. The following resources for replacing copyrighted map figures may be helpful:

NASA Earth Observatory (public domain): http://earthobservatory.nasa.gov/Landsat:
http://landsat.visibleearth.nasa.gov/

USGS EROS (Earth Resources Observatory and Science (EROS) Center) (public domain): http://eros.usgs.gov/#Natural Earth (public domain): http://www.naturalearthdata.com/

- Figure 1.B. contains a view of the Crop Layer Data as seen on the CropScape website: https://nassgeodata.gmu.edu/CropScape/. These data are considered public domain and are freely available. As such, no permission is required to use and publish these data. Moreover, the source of these data was cited in our manuscript.

- Figure 2 contains our results, plotted over the state boundaries (which are also public domain).

- Done

Response to comments from review #1

• Line 23: “Widespread declines in wild bee populations necessitates”—either remove the “s” from “declines” or from “necessitates” to correct the subject-verb agreement.

- Done

• Line 47: A closing parenthesis appears to be missing.

- Replaced ‘(e.g., (8,9)’ by ‘(e.g., 8,9)’.

• Line 55: There is a double space between “them.” and “Consequently.”

- Removed the extra spaces.

• Lines 80–82: The distinction between ‘environmental health’ and ‘environmental quality’ is unclear. Are there rigorous definitions that differentiate these terms? Additionally, the statement that invertebrates are better proxies for one and birds for the other requires more explanation or supporting references.

- We agree that the two terms are often used interchangeably and have been more specific that insects are more often used to indicate environmental health as related to pollution and other abiotic factors, whereas birds are more regularly used to assess habitat quality or condition. These distinctions are mentioned in 2 of the 4 papers cited.

• Lines 86–93: These lines, discussing the use of single species or assemblages, directly relate to concepts introduced in lines 74–76. Consider merging lines 74–76 with lines 86–93 for a more cohesive discussion, and then proceed with lines 76–86, which focus on specific taxa and their relevance.

- We updated the order of the information.

• Lines 113–114: There seems to be another missing parenthesis in this section.

- We removed the parenthesis before the citation “scales (e.g., microhabitat, stand, landscape, and region; 43,72,73)”, thus avoiding the need to add a second parenthesis after the citation.

• Lines 149–151: I am not sure how bee richness was calculated. Specifically, how did you integrate data from net sampling when averaging the number of species per trap per survey? Further clarification on this methodology would be helpful.

- The Bee Inventory and Monitoring Laboratory Protocol recommends and typically uses both sampling methods at each site surveyed. As such, we assumed both techniques were used to sample the species at each site and did not further account for protocol used to sample the bees.

• Lines 164–165: This might be a naïve observation, but in this semi-structured dataset, surveys appear to follow different protocols. Couldn’t this lead to significant variations in the number of species per survey, not due to actual differences in species richness, but as a result of methodological discrepancies? For instance, depending on the sampling method (e.g., nets, traps, or nets + traps) or the sampling duration, the number of species recorded at the same site on the same day can vary substantially; it can easily double or triple. How do you think this variability might influence the trends observed in your study? Additionally, why did you use the mean instead of the median to summarise species richness? Wouldn’t the median reduce the influence of outliers? Alternatively, why not consider the total number of species observed across all surveys within a grid cell?

- We agree that combining various protocols in one dataset might introduce bias in the number of bee species detected. However, the bee data in the semi-structured dataset rarely included information about what protocol was used (see paper Rousseau et al. 2024), making it difficult to select or account for each protocol. This is one reason we completed the second analysis with a structured dataset from a data owner from which we knew the protocol used.

- We used mean instead of median, because it is the most commonly used summary statistic and because we did not observe major sources of outliers.

- We found that including data from surveys with only one bee introduced a spatial bias in our study area because these one-bee surveys were typically from iNaturalist, which has most of its records in highly populated areas.

• Line 170: Why did you not include pre-nuptial migration periods, such as April, in your bird-based index? April is a critical period for many early-season bee species, and excluding it might overlook their habitat requirements. Including data from pre-nuptial migration sites and their associated bird diversity could potentially improve the index's applicability to early-season bees. While, of course, re-extracting bird data is beyond the scope of this study, do you think this aspect is worth exploring in future research?

- While we included bee data from all months (including April), we excluded birds from April because most migratory species are still in transit and present in locations where they do not breed. Migrating birds are expected to be less reliable indicators of habitat conditions because (1) they are generally known to be more flexible (less specific) in habitat use and, in many cases, (2) might only be at a given location for a few hours (Stanley et al. 2021. Ecosphere). In our analysis, these birds would introduce an additional bias.

• Line 333: It seems that a track change was not removed, specifically in the space between “bees” and (101).

- Thank you; we removed the track change.

• Figure 2: In the figure legend, could you clarify what “scaled uncertainty” represents? Is it expressed as a percentage or as a number of species?

- We added information about scale uncertainty in the caption.

• Figure 2: It might be valuable to include a map showing actual bee richness based on existing data. This would allow for a direct comparison between observed richness and the predicted values, providing additional context and insight into the model's accuracy.

- Indeed, we considered adding a map of the actual bee richness in the supplementary material. However, the bee data are very sparse geographically (at least in most areas) and the grid cells are very small. As such we could not create a map that aesthetically represented actual bee richness.

Response to comments from review #2

• The statement (lines 99 – 101) “however, to our knowledge, González et al. (17) and Fleishman et al. (27) were first to document that a combination of environmental variables and indicator species best explained restoration success and variation in species richness, respectively”, is a bit confusing. Since it is unlikely that these are the only that have adopted this approach (as one might think), it would be convenient to cite at this point some additional references published after them that can be considered background to the present research.

- This was surprising to us as well. After much research, these are the two publications we could find. Please let us know if we missed additional publications known to you.

• If anything, in relation to the methodology, I would appreciate an explanation in some detail of the methodological basis of the eBird product cited (reference 92, Fink et al., 2021: Lines 250-252 “we needed estimates of relative bird abundance in all grid cells, not only those with eBird checklists, therefore we used the estimated relative abundance per species per grid cell from eBird Status Data Products (92)”. Although this product is explained in the link associated to the reference, it seems appropriate that the authors (and not the reader) be the ones to extract -and include in the text- this necessary methodological note.

- We added a short methodological note and referred to a paper (in addition to citing the dataset).

• Lines 350-350: The sentence "Fourth, the combined bird and land cover variables selected as predictors represent a wide range of habitats, which suggest that a higher bee richness may be associated with heterogeneous landscapes (110–112) that include multiple types of nesting substrates to accommodate ground and cavity nesters and a diversity off lower resources, from crops, shrubs, and trees", is also a bit confusing since the argument combines nesting substrates(¿for birds?) and trophic resources (for bees? for both?). The relationship of birds with heterogeneous landscapes and structurally diverse habitats, and their consequent indicator value for habitat diversity for pollinators, should probably be differentiated. Likewise, it should be accompanied by some references that support the first relationship, since all those cited (110-112) refer exclusively to the habitat relationships of pollinators.

On the contrary, in the sentence between lines 392-394, which refers entirely to pollinators, "The structured dataset included protocol and effort information, enabling us to generate more precise bee richness estimates across space (121,122)", the second reference cited is on birds. It is not clear to me what justifies its use.

- We referred to bee habitats, including bee nesting substrates and resources, which we clarified.

- The second reference (122, now 123), which happens to be about birds, shows how the inclusion of protocol and effort helps generate more precise estimates.

• The article goes into little detail about the cautions to be taken when choosing birds as surrogate for invertebrates (wild pollinators in this case), especially taking into account the extensive literature on the lack of correspondence between protected areas designated with criteria based on each of these taxocenoses. For example, the following works or others with a similar approach can be cited:

o Guareschi, S., Abellán, P., Laini, A., Green, A. J., Sánchez-Zapata, J. A., Velasco, J., & Millán, A. (2015). Cross-taxon congruence in wetlands: assessing the value of waterbirds as surrogates of macroinvertebrate biodiversity in Mediterranean Ramsar sites. Ecological Indicators, 49, 204-215.

o Oberprieler, S. K., Andersen, A. N., Gillespie, G. R., & Einoder, L. D. (2019). Vertebrates are poor umbrellas for invertebrates: cross‐taxon congruence in an Australian tropical savanna. Ecosphere, 10(6), e02755.

o Rooney, R. C., & Bayley, S. E. (2012). Community congruence of plants, invertebrates and birds in natural and constructed shallow open-water wetlands: do

---

## [Decision Letter · Decision Letter 1]

7 Mar 2025

Where the wild bees are: birds improve indicators of bee richness

PONE-D-24-49004R1

Dear Dr. Rousseau,

We’re pleased to inform you that your manuscript has been judged scientifically suitable for publication and will be formally accepted for publication once it meets all outstanding technical requirements.

Kind regards,

Vicente Martínez López

Academic Editor

PLOS ONE

Additional Editor Comments (optional):

Reviewers' comments:

Reviewer's Responses to Questions

**Comments to the Author**

1. If the authors have adequately addressed your comments raised in a previous round of review and you feel that this manuscript is now acceptable for publication, you may indicate that here to bypass the “Comments to the Author” section, enter your conflict of interest statement in the “Confidential to Editor” section, and submit your "Accept" recommendation.

Reviewer #1: All comments have been addressed

Reviewer #2: All comments have been addressed

2. Is the manuscript technically sound, and do the data support the conclusions?

Reviewer #1: Yes

Reviewer #2: Yes

3. Has the statistical analysis been performed appropriately and rigorously? 

Reviewer #1: Yes

Reviewer #2: Yes

4. Have the authors made all data underlying the findings in their manuscript fully available?

Reviewer #1: Yes

Reviewer #2: Yes

5. Is the manuscript presented in an intelligible fashion and written in standard English?

Reviewer #1: Yes

Reviewer #2: Yes

6. Review Comments to the Author

Reviewer #1: Dear Authors,

Thank you for your revision. I have no additional comments and fully support the acceptance of this paper.

Best regards,

Maxence Gérard

Reviewer #2: (No Response)

7. PLOS authors have the option to publish the peer review history of their article (what does this mean? ). If published, this will include your full peer review and any attached files.

**Do you want your identity to be public for this peer review?** For information about this choice, including consent withdrawal, please see our Privacy Policy .

Reviewer #1: **Yes: ** Maxence Gérard

Reviewer #2: No

---

## [Editor Report · Acceptance letter]

PONE-D-24-49004R1

PLOS ONE

Dear Dr. Rousseau,

I'm pleased to inform you that your manuscript has been deemed suitable for publication in PLOS ONE. Congratulations! Your manuscript is now being handed over to our production team.

Kind regards,

on behalf of

Dr. Vicente Martínez López

Academic Editor

PLOS ONE